# Interactions of Nereistoxin and Its Analogs with Vertebrate Nicotinic Acetylcholine Receptors and Molluscan ACh Binding Proteins

**DOI:** 10.3390/md20010049

**Published:** 2022-01-04

**Authors:** William R. Kem, Kristin Andrud, Galen Bruno, Hong Xing, Ferenc Soti, Todd T. Talley, Palmer Taylor

**Affiliations:** 1Department of Pharmacology and Therapeutics, College of Medicine, University of Florida, 1200 Newell Drive, Gainesville, FL 32610, USA; Kristin.Andrud@du.edu (K.A.); galen.bruno@yahoo.com (G.B.); hong.xing@ufl.edu (H.X.); 2Department of Neurology, College of Medicine, University of Florida, 1200 Newell Drive, Gainesville, FL 32610, USA; 3Department of Pharmacology, Skaggs School of Pharmacy and Pharmaceutical Sciences, and School of Medicine, University of California, San Diego, La Jolla, CA 92093, USA; tttalley@live.com (T.T.T.); pwtaylor@health.ucsd.edu (P.T.)

**Keywords:** acetylcholine binding protein, α-bungarotoxin, annelid, cholinergic, insecticide, nereistoxin, nicotinic acetylcholine receptor, toxin

## Abstract

Nereistoxin (NTX) is a marine toxin isolated from an annelid worm that lives along the coasts of Japan. Its insecticidal properties were discovered decades ago and this stimulated the development of a variety of insecticides such as Cartap that are readily transformed into NTX. One unusual feature of NTX is that it is a small cyclic molecule that contains a disulfide bond. In spite of its size, it acts as an antagonist at insect and mammalian nicotinic acetylcholine receptors (nAChRs). The functional importance of the disulfide bond was assessed by determining the effects of inserting a methylene group between the two sulfur atoms, creating dimethylaminodithiane (DMA-DT). We also assessed the effect of methylating the NTX and DMA-DT dimethylamino groups on binding to three vertebrate nAChRs. Radioligand receptor binding experiments were carried out using washed membranes from rat brain and fish (*Torpedo*) electric organ; [^3^H]-cytisine displacement was used to assess binding to the predominantly high affinity alpha4beta2 nAChRs and [^125^I]-alpha-bungarotoxin displacement was used to measure binding of NTX and analogs to the alpha7 and skeletal muscle type nAChRs. While the two quaternary nitrogen analogs, relative to their respective tertiary amines, displayed lower α4β2 nAChR binding affinities, both displayed much higher affinities for the *Torpedo* muscle nAChR and rat alpha7 brain receptors than their respective tertiary amine forms. The binding affinities of DMA-DT for the three nAChRs were lower than those of NTX and MeNTX. An AChBP mutant lacking the C loop disulfide bond that would potentially react with the NTX disulfide bond displayed an NTX affinity very similar to the parent AChBP. Inhibition of [^3^H]-epibatidine binding to the AChBPs was not affected by exposure to NTX or MeNTX for up to 24 hr prior to addition of the radioligand. Thus, the disulfide bond of NTX is not required to react with the vicinal disulfide in the AChBP C loop for inhibition of [^3^H]-epibatidine binding. However, a reversible disulfide interchange reaction of NTX with nAChRs might still occur, especially under reducing conditions. Labeled MeNTX, because it can be readily prepared with high specific radioactivity and possesses relatively high affinity for the nAChR-rich *Torpedo* nAChR, would be a useful probe to detect and identify any nereistoxin adducts.

## 1. Introduction

Over 80 years ago, a neurotoxin was isolated from a large (~40 cm) annelid worm (*Lumbriconereis heteropoda*) that occurs along the coasts of Japan. The discovery of this toxin resulted from use of the worm as a fish bait. It was noticed by fisherman that flies eating the dead worms would often become paralyzed and die. Fishermen who had handled this annelid worm would occasionally develop a headache, nausea, and respiratory difficulties. Nereistoxin (NTX, Figure 1), the active substance, was isolated by Nitta [1] and 28 years later its structure was reported by Okaichi and Hashimoto [2]. It was found to be a nicotinic acetylcholine receptor (nAChR) antagonist when applied to frog skeletal muscle [3]. Under some conditions, when applied to mammalian skeletal muscle, NTX also initially caused a small, transient membrane depolarization [4]. Administration of a relatively low dose of NTX to experimental mammals caused respiratory depression and at higher doses central nervous system effects [5]. NTX possesses a disulfide bond within its five-membered dithiolane ring. Its interaction with retinal ganglia nAChRs under reducing conditions was consistent with formation of an intermolecular disulfide bond with a vicinal disulfide occurring within the C-loop of the nAChR α subunit that is part of the ACh binding site [6,7,8].

NTX is relatively toxic to insects, especially lepidopteran larvae such as that of the rice stem boring insect. Consequently, numerous analogs of NTX including cartap (Figure 1) have been used as insecticides. The myriad nAChRs of insects are primarily expressed within their central nervous systems. NTX suppresses electrical signaling within the insect central nervous system and acts as an antagonist at insect nAChRs [9]. Difficulties in the expression of insect nAChRs in cultured cells have slowed scientific analysis of the mechanism of NTX action on these insect nAChRs. More readily expressed chimeric nAChRs constructed with insect and chicken neuronal nAChR DNA sequences were blocked by NTX in a manner suggesting non-competitive antagonism; this was interpreted as being consistent with ion channel blockade [10]. A more recent investigation also suggested that NTX may largely exert its insecticidal action by blocking insect nAChR ion channels [11].

Vertebrate skeletal muscle neuromuscular junction and autonomic ganglionic nAChRs are essential for synaptic transmission. At the frog neuromuscular junction and the related marine ray *Torpedo* electric organ, the site most sensitive to NTX blockade is the ACh binding site rather than the ion channel [3,4]. While the effects of NTX on mammalian skeletal muscle nAChRs have been reported, little has been published concerning its direct actions on mammalian brain nAChRs. A primary goal of this study was to assess NTX interaction with α4β2 and α7 nAChRs, the two major subtypes expressed in the brain.

The molecular mechanism(s) by which NTX acts as an antagonist at vertebrate nAChRs is not clear. NTX is much smaller and more flexible than most nAChR antagonists, such as dihydro-β-erythroidine, methyllycaconitine, and peptide toxins from elapid and marine snails. In addition to investigating N-methylnereistoxin (MeNTX) to assess the importance of ionization and increased bulkiness of the amino group, we also investigated two NTX analogs (5-dimethylamino-1,3-dithiane (DMA-DT) and 5-trimethylamino-1,3-dithiane (TMA-DT)) which have a methylene group inserted between the two NTX sulfur atoms and therefore lack the disulfide bond (Figure 2). They were used to assess whether the presence of the potentially reactive disulfide bond in NTX is necessary for its activity.

## 2. Results

### 2.1. Effects of NTX on α4β2 Neuronal Receptors

First, the binding properties of NTX and its analogs with rat brain α4β2 nAChRs were considered. We focused on measurements of binding affinity for these receptors, since it has already been demonstrated that NTX acts mainly as an antagonist [3,4,5]. NTX displayed its highest affinity (IC_50_ = 60 µM) for the α4β2 subtype, which is also very sensitive to nicotine relative to the other nAChRs investigated here. Methylation of the NTX dimethylamino group, producing MeNTX, led to a small decrease in affinity (increased IC_50_) for this receptor (Figure 3 and Table 1).

The effects of quaternizing the dimethylamino group of nereistoxin were similar to what is observed for this nAChR subtype when nicotine is methylated at the equivalent pyrrolidinyl ring nitrogen [12]. The calculated equilibrium inhibition constant for NTX binding was approximately 1,000× higher than the IC_50_ of nicotine at this receptor, using the same experimental conditions. NTX and MeNTX inhibited ACh activation of human α4β2 nAChRs expressed in a cultured cell line (Figure 4).

Using a cell line expressing human α4β2 nAChRs and the sensitive FlexStation assay (see Methods), we failed to detect a membrane depolarizing action of MeNTX that was reported for NTX (Figure 5). Even at 300 µM concentration, MeNTX failed to activate this nAChR.

### 2.2. Effects on α7Neuronal Receptors

MeNTX had an approximately 9-fold greater affinity for the α7 receptor than did NTX, in contrast with the lower potency of MeNTX relative to NTX observed in the α4β2 binding experiments (Figure 6). While TMA-DT was not as good an antagonist as MeNTX, its higher binding affinity relative to DMA-DT was also likely due to its cationic property (Table 1).

### 2.3. Torpedo Electric Organ nAChR Binding 

MeNTX had an approximately 20-fold greater affinity for this muscle receptor relative to NTX. This was the greatest increase in binding affinity observed in our analysis of the three different nAChRs. Disruption of the disulfide bond in TMA-DT reduced affinity relative to NTX. If NTX binds to this receptor in an essentially irreversible manner, it would be expected that sensitivity to NTX inhibition would be related to NTX pre-incubation time. Our initial binding data for 30 min preincubation with NTX (Figure 7, Left) was similar to data from simultaneous addition of NTX with [^125^I-BTX (Figure 7, Right). MeNTX was much more potent in inhibiting BTX binding, regardless of whether it was added 30 min before BTX or concomitantly.

### 2.4. Binding of NTX and MeNTX to Molluscan ACh Binding Proteins

Since the molluscan AChBPs contain very similar ACh binding sites to those of nAChRs, we measured the abilities of the nereistoxins to inhibit the binding of the potent agonist [3H]-epibatidine (Figure 8, Table 2). While the *Lymnaea stagnalis* AChBP binding affinity of MeNTX was almost the same as for NTX, the MeNTX affinity of *Ls* C187S mutant (data in Table 2 legend), was ~4-fold greater than for NTX. The affinity of *Aplysia californica* AChBP for MeNTX was ~3.5-fold less than for NTX. However, the affinity of *Ac* AChBP Y55W mutant for MeNTX was greatly enhanced relative to that for NTX. While the binding affinities of the nereistoxins for the AChBPs varied, they were in the same concentration range as for the nAChRs.

## 3. Discussion

The three major goals of this investigation were to assess: (1) the affinities of NTX for the major mammalian brain (α4β2 and α7) and marine ray electric organ (*Torpedo californica*) nAChRs, (2) the effect of quaternization of the dimethylamino group, and (3) the effect of inserting a methylene spacer moiety between the two sulfur atoms.

We compare the IC_50_ estimates for each compound and receptor rather than K_i_ estimates, since we have not demonstrated that binding is reversible and competitive. The use of the Cheng-Prusoff equation to derive K_i_’s from IC_50_s assumes reversible competition. Delpech et al. [13] actually found that increasing NTX concentrations progressively reduced the peak responses of their chimeric chicken-insect nAChRs, even at high ACh concentrations. They interpreted these results as indicating a non-competitive antagonism by NTX. However, these results could also be obtained if NTX irreversibly binds to the ACh sites. In fact, they indicated that the NTX inhibition was irreversible at high concentrations or after prolonged exposure.

Relative to the other two nAChRs, the α4β2 subtype displayed the highest affinity for NTX. N-methylation of NTX, creating the permanently charged analog MeNTX reduced affinity for this nAChR. The NTX dimethylamino pKa has been reported to be 7.2 [14]. Since it has been shown that the ionized form of nicotine is the active form on rat [15], amphibian [16] and ray [17] electric organ nAChRs, one would anticipate that the cationic forms of NTX and DMA-DT toxins would have much greater affinity for the ACh binding site. The quaternary methyl analogs are entirely cationic in the physiological pH range and would be expected to bind more tightly than the unprotonated molecule due to electrostatic interaction with the electronegative environment of the ACh binding site. Methylation of the 1’ tertiary amino group N in the pyrrolidinyl ring in nicotine reduces its affinity for the α4β2 receptor but not the α7 receptor [12]. Most NTX and DMA-DT molecules would be predicted to be non-ionized in our experiments, which were carried out at pH 7.4, slightly above the pKa of nereistoxin and presumably DMA-DT. Since the brain α7 and muscle nAChRs displayed much higher (8- and 22-fold, respectively) affinities for binding MeNTX, one can tentatively conclude that the added methyl group enhances the cation-pi bonding interaction of these compounds with these nAChRs.

In the absence of additional data, the lower affinity of the DMA-DT relative to NTX for all three nAChRs can be interpreted several ways. The greater size of its ring (6 instead of 5 membered) may be a major factor, as the well-known cation-pi “cage” may not readily accommodate the larger size. Another possibility is that a disulfide structure is optimal. 

Like MeNTX, TMA-DT displayed higher affinity than DMA-DT for the two α-bungarotoxin binding receptors. This was consistent with the relatively high affinity of 1’-methylnicotinium binding to rat brain α7 nAChRs (12). 

An extensive literature exists regarding the reactivity of the vicinal disulfide in the C loop [6,7,8,18,19,20]. Of the three disulfide bonds within the α-subunit, only this vicinal disulfide is readily reduced by dithiothreitol under physiological conditions [18]. It is now thought that the vicinal Cys192-Cys193 disulfide bond is a key contributor to an extensive system of H-bonding interactions that stabilize the active receptor conformation in the ACh binding region [19]. The Cohen laboratory has studied the ability of quaternary ammonium compounds tethered to the vicinal thiol groups to act as agonists or antagonists at the *Torpedo* nAChR. They found that shorter choline-like tethered compounds tended to be antagonists, but once the length of the molecule was more like in ACh the compound acted as an agonist. Interestingly, a compound with the length expected to possess a distance similar to the distance between the dimethylamino group of NTX and one of its sulfurs acted like an antagonist [20].

The molluscan AChBPs have been extremely useful for visualizing what likely happens at the ACh binding sites in nAChRs due to their very similar structure and the high resolution of their crystal structures [21,22]. In general, small ligands for this site allow closure of the C loop and a general closure of the other loops around the ligand as well, which allows activation of the receptor and the opening of its ion channel. On the other hand, larger ligands tend to be partial agonists or full antagonists depending on how much they allow the C loop to cap the ligand [22]. NTX is roughly the size of ACh, so it is surprising that it is an antagonist. The binding experiments with the molluscan AChBPs failed to detect any effect of 24 hr preincubation of NTX and MeNTX on their inhibition of [3H]-epibatidine binding. The toxins also inhibited epibatidine binding to the *L. stagnalis* BP C187S mutant (See Table 2 legend for this data), which lacks one of the sulfurs required for formation of the C loop disulfide bond. Overall, although there were differences between Ls and Ac BP relative affinities for NTX and MeNTX, the BPs displayed binding affinities for NTX and MeNTX that were similar to those observed for the three vertebrate nAChRs. It was interesting that the *Ac* BP Y55W mutant displayed a higher affinity for MeNTX than NTX, in contrast with *Ac* BP. This mutant was made and tested because it more closely resembles vertebrate α7 nAChRs, which have a tryptophan at the homologous position.

Our results with dithianes DMA-DT and TMA-DT suggest NTX action does not require disulfide bond interchange and covalent bond formation between the NTX and loop C vicinal disulfide bond. Nevertheless, it is possible that such a disulfide interchange mechanism may be facultative (and primarily occur under reducing conditions) rather than being essential, i.e., that such an exchange contributes to the stability of the NTX-receptor complex but is not necessary for the antagonist activity of NTX. We suggest that the most direct test of the disulfide interchange hypothesis of NTX action would be to measure irreversible binding of radiolabeled MeNTX to *Torpedo* nAChRs. Such a radioligand could be easily prepared by methylation of NTX using a tritiated methylhalide. After exposure to the radiolabeled MeNTX, the membrane preparation would be subjected to detergent solubilization, proteolytic or cyanogen bromide cleavage and SDS-gel electrophoresis to isolate and characterize the reaction sites.

We have not yet deciphered the mechanism by which this small molecule can be an nAChR antagonist. We suggest that NTX and its analogs are affecting the binding of the radioligands by direct competition for the ACh binding sites, as NTX displays very low affinity for the phencyclidine binding site within the electric organ muscle nAChR ion channel [23]. This is also based on data demonstrating that binding to the nAChR ion channel site where phencyclidine, histrionicotoxin and some other channel blockers bind does not inhibit binding of α-BTX at the ACh binding sites [24]. We did observe that NTX inhibited [^3^H]-TCP (radiolabelled analog of phencyclidine) binding to the *Torpedo* membranes, but this only occurred at NTX concentrations that were ~10X higher than were required to inhibit α-BTX binding (results not shown). Since most competitive antagonists are relatively large rigid molecules, elucidating the molecular mechanism by which NTX, an extremely small molecule, acts as an nAChR antagonist merits further investigation.

## 4. Materials and Methods

### 4.1. NTX Compounds

NTX oxalate was a gift from Takeda Chemical Industries, Ltd. (See Acknowledgements) and was also purchased from Wako Chemical Company. MeNTX (Table 1) was prepared by reacting methyl iodide (3-fold molar excess) with NTX free base; the product was isolated as the hydrogen iodide salt. The NTX analog 5-dimethylamino-dithiane (DMA-DT, Table 1) was prepared from 1,3-dithian-5-one [14,25]. The MeNTX analog 5-trimethylamino-1,3-dithiane hydrogen iodide (TMA-DT) was prepared by reacting methyl iodide with DMA-DT free base, as was MeNTX prepared from NTX. NMR and MS analyses of the three NTX analogs were in agreement with the published data. The purities of NTX and its three analogs were assessed by C18RP HPLC with 10% acetonitrile −90% water −0.1% TFA development.

### 4.2. Radioligand Binding Assays

[^3^H]-Cytisine (Perkin-Elmer) and [^125^I]-α-BTX (General Electric) were used in binding experiments with whole rat brain and *Torpedo* electric organ membranes prepared according to procedures described in detail elsewhere [12,26,27,28,29]. Rat brain membranes (100 μg of protein) expressing human α4β2 nAChRs were incubated with 0.5 nM [^3^H]-cytisine in a final volume of 500 µL binding saline for 4 h at 5 °C. The experiments with α7 nAChRs rat brain membranes involved incubation with 0.2 nM [^125^I]-α-Btx for 3 h at 37 °C to assure that equilibrium was reached. Ten different concentrations of the experimental compound were usually tested in quadruplicate. Nonspecific binding was measured in the presence of 1 mM (S)-nicotine hydrogen tartrate (Sigma-Aldrich). Data were fitted using GraphPad Prism software (Version 4 GraphPad Software, San Diego, CA, USA) by nonlinear regression analyses to a sigmoidal one-site model with variable slope. Compound affinity for the *Torpedo californica* electric organ membrane nAChR was assessed by inhibition of 0.5 nM [^125^I]-α-BTX binding to 20 µg washed membrane/tube over a 3 hr incubation period at 25 °C.

### 4.3. Radioligand Binding Assay for AChBPs

The AChBP proteins were expressed and purified according to Hansen et al. [30]. Determination of compound affinity for an AChBP utilized a modification of the Scintillation Proximity Assay reported previously [31,32]. Briefly, the AChBP constructs (0.5–1.0 nM final concentration of binding sites) were combined with polyvinyltoluene anti-mouse SPA scintillation beads (0.17 mg/mL final concentration, Perkin Elmer), monoclonal anti-FLAG M2 antibody from mouse 1:8000 dilution (Sigma), and 0.1 M NaPO_4_ buffer, pH 7.0. This cocktail was distributed to a 96-well plate and incubated with either vehicle (total binding), unknown sample (10 µM final concentration), or a saturating concentration (10 µM) of the known competitive ligand methyllycaconitine to determine the nonspecific binding. The radioligand (±)-[^3^H]-epibatidine (Perkin Elmer, 5 nM final concentration) was added to each well with varying the concentrations of competing ligand and allowed to incubate at room temperature for 1 hr. Plates were read on a 1450 MicroBeta TriLux liquid scintillation counter (Wallac) and the output normalized. All measurements were conducted in duplicate a minimum of 3 times. The IC_50_ values were calculated using GraphPad Prism version 4.02 (GraphPad Software, 2006, San Diego, CA, USA, www.graphpad.com).

### 4.4. Nicotinic Receptor FlexStation Functional Assays

TsA201 cells expressing human α4β2 were maintained in media consisting of Dulbecco’s Modified Eagle medium supplemented with 10% FBS, 100 units/mL penicillin and 100 μg/mL streptomycin, 2 mM L-glutamine, 0.5 mg/mL zeocin, and 0.6 mg/mL geneticin. Cells were grown in 75 cm^2^ culture flasks, which were housed in a humidified incubator (Fisher Scientific, Atlanta, GA, USA at 37 °C in an atmosphere of 5% CO_2_. They were grown to around 80–90% confluence after harvesting with 0.25% trypsin and being split weekly at a subcultivation ratio of between 1:6 and 1:10.

Our experimental protocol was based on the initial study of Fitch et al. [33]. Cells were seeded at a density of roughly 5 × 10^4^ to 10^5^ cells/well in 96-well flat-bottom black wall culture plates coated with 50 μg /mL poly-D-lysine hydrobromide (Sigma-Aldrich, 70–150 kDa) and grown overnight in 100 μL culture medium. A proprietary membrane potential dye obtained from Molecular Devices (San Diego, CA, USA) was prepared by dissolving one bottle of dye into 30 mL of Hanks Saline (pH = 7.4) containing 20 mM HEPES buffer. The cells were incubated with 100 μL of dye for 30 min at 37 °C prior to the robotically controlled concentration-response experiment. Serial dilutions of a compound for dose-response analysis were prepared in 96-well plates by evaporation of a methanolic stock solution and then reconstituted in the appropriate volume of Hanks saline. Fluid transfer and readings were performed by a FlexStation fluorimeter (Molecular Devices). Excitation and emission wavelengths were set to 530 nm and 565 nm with a cutoff of 550 nm. The first 17 s were used as a basal reading. At 18 s, a test compound was added to determine the EC_50_, followed by addition of 25 μL KCl (40 mM final concentration) at 160 s to serve as a fluorescence calibrant. NTX displayed no measurable depolarizing activity on the TsA201 cells and was then tested for its ability to inhibit a 5-µM control ACh response.

## Figures and Tables

**Figure 1 marinedrugs-20-00049-f001:**
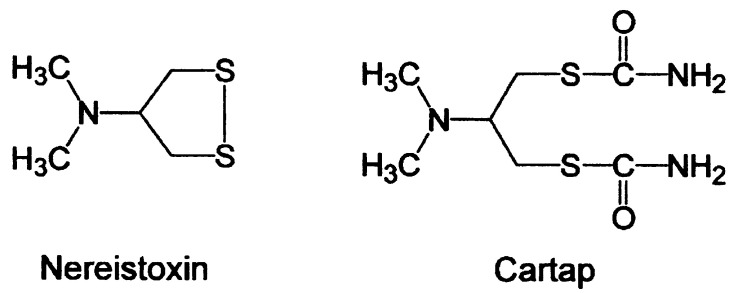
Structures of nereistoxin (NTX, 4-dimethylamino-1,2-dithiolane) and cartap, one of the nereistoxin-based insecticides. Cartap primarily acts as a pro-insecticide, being converted into the more active NTX.

**Figure 2 marinedrugs-20-00049-f002:**
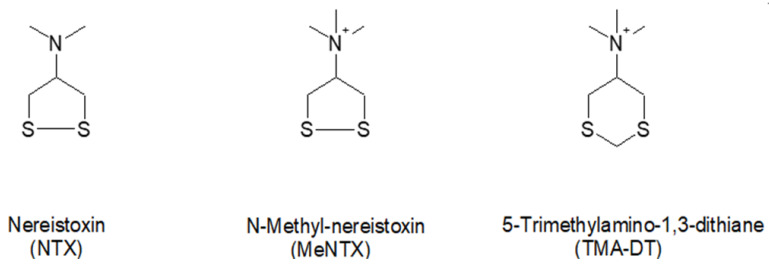
Comparison of the structures of the N-quaternized analogs with nereistoxin. The ring of the tertiary amine dithiane analog MA-DT is identical with that of TMA-DT.

**Figure 3 marinedrugs-20-00049-f003:**
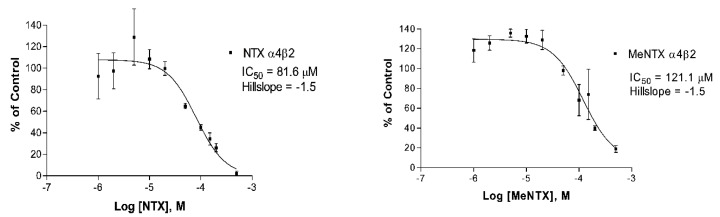
Equilibrium [^3^H]-cytisine displacement experiments measuring affinity of NTX (**Left**) and MeNTX (**Right**) for rat brain α4β2 nAChRs. These two curves are from individual experiments, each using 48 tubes containing equal amounts of membrane but measuring binding at 10 different concentrations of NTX or MeNTX, as well as total binding and non-specific binding. All 12 conditions were measured in quadruplicate. Standard error bars are shown for each concentration.

**Figure 4 marinedrugs-20-00049-f004:**
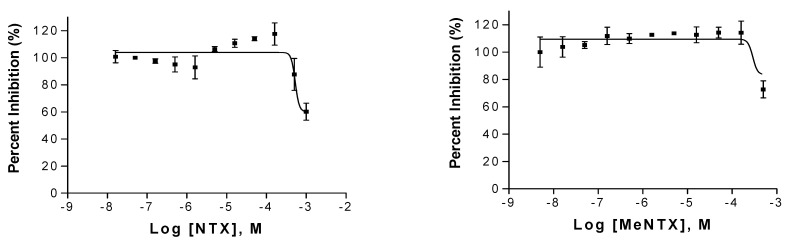
Inhibition of human α4β2 nAChR mediated cell response to 5 μM acetylcholine by NTX (**Left**) and MeNTX (**Right**). The receptor was expressed in TsA201 cells and responses were measured in a FlexStation using a flourescent dye sensing membrane potential. Standard error bars are shown for each concentration.

**Figure 5 marinedrugs-20-00049-f005:**
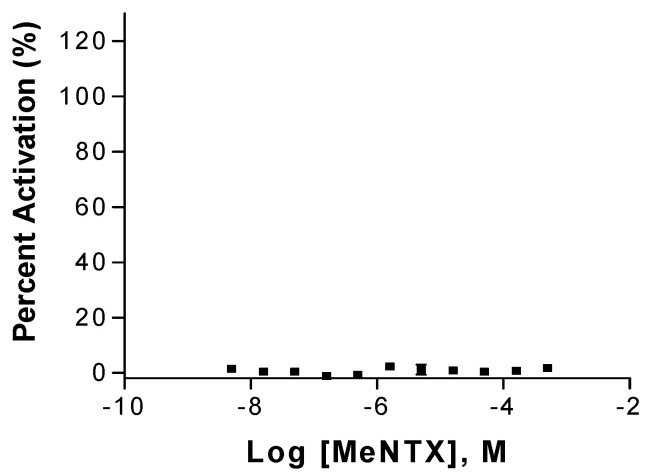
Failure of MeNTX to activate TsA201 cells expressing human α4β2 nAChRs. Activation is expressed relative to stimulation by 5 µM ACh.

**Figure 6 marinedrugs-20-00049-f006:**
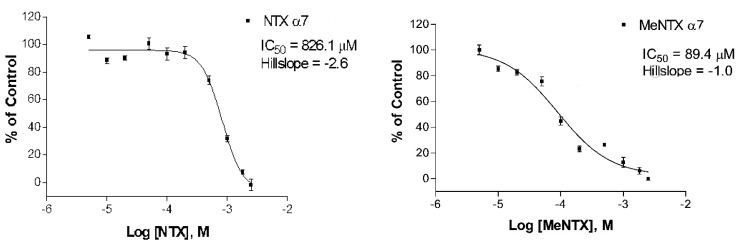
A representative [^125^I]-αBTX binding displacement experiment measuring affinity of NTX (**Left**) and MeNTX (**Right**) interaction with the predominant homomeric α7 nAChR. MeNTX displayed approximately 10X higher affinity than NTX at this nAChR. Each displacement curve is from a single experiment; average estimates obtained from several identical experiments are found in Table 1.

**Figure 7 marinedrugs-20-00049-f007:**
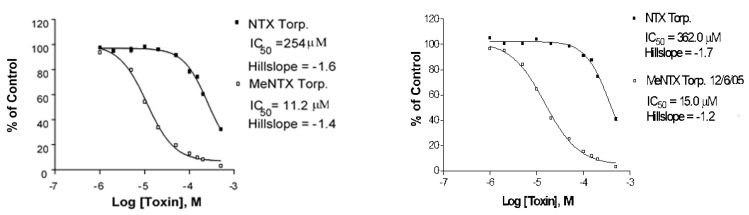
Single [^125^I]-αBTX binding displacement experiments measuring affinity of NTX and MeNTX interaction with the *Torpedo* fish electric organ fetal type neuromuscular nAChR. *(***Left)**: 30 min preincubation with NTX; (**Right**): Simultaneous addition of NTX with BTX. Additional data from other binding experiments were used to obtain the average IC_50_ estimates in Table 1.

**Figure 8 marinedrugs-20-00049-f008:**
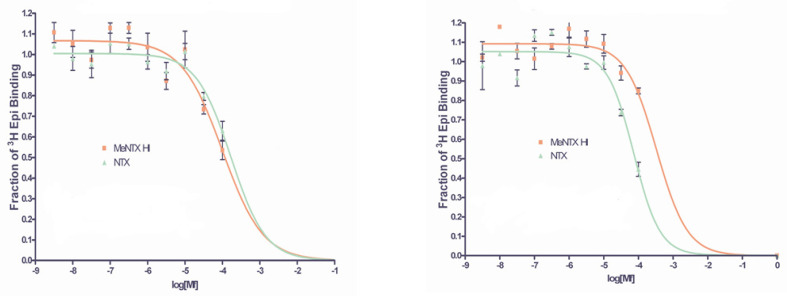
Inhibition of AChBP [^3^H]-epibatidine binding by NTX and MeNTX. Two different molluscan AChBPs were tested. **Left**, *Lymnaea stagnalis* AChBP; **Right**, *Aplysia californica* BP. These are representative results from single experiments. Additional experiments were carried out to obtain the average IC_50_ estimates shown in Table 2.

**Table 1 marinedrugs-20-00049-t001:** Median inhibitory concentrations for NTX compound inhibition of radioligand binding to the two major mammalian brain nAChRs and to *Torpedo* muscle nAChRs. IC_50_ estimates (µM) are accompanied by a standard error (SEM). The number of experiments (n) is noted within parentheses. In each experiment, ten different concentrations of each compound were assessed in quadruplicate. Total binding in absence of the nereistoxin compound and non-specific binding were also assayed in quadruplicate. ^1^ Determined by displacement of [^3^H]-cytisine; ^2^ Determined by displacement of [^125^I]-α-BTX.

	Rat Brain	Rat Brain	*Torpedo*
Compound	α4β2 ^1^	α7 ^2^	Electric Organ ^2^
		IC_50_ ± SEM (µM)	
NTX	60 ± 22 (6)	390 ± 8.6 (4)	230 ± 71 (3)
MeNTX	120 ± 13 (3)	45 ± 2.3 (2)	9.3 ± 4.9 (4)
DMA-DT	160 ± 96 (3)	640 ± 220 (2)	>500 (1)
TMA-DT	290 ± 60 (4)	90 ± 4.3 (2)	150 (1)

**Table 2 marinedrugs-20-00049-t002:** NTX compound inhibition of [^3^H]-epibatidine binding to molluscan acetylcholine binding proteins. IC_50_ estimates (µM) are accompanied by a standard error (SEM). The number of experiments (n) is noted within parentheses. In each experiment, at least eleven compound concentrations were assessed in quadruplicate. The *L. stagnalis* C187S mutant NTX IC_50_ = 160 ± 42 (3) µM and MeNTX IC_50_ = 41 ± 20 (3) µM.

Compound	*Ls* BP	*Ac* BP	*Ac* BP Y55W
		IC_50_ ± SEM (µM)	
NTX	140 ± 24 (2)	71 ±1.9 (3)	110 ± 25 (3)
MeNTX	163 ± 8.7 (3)	260 ± 71 (3)	74 ± 5.2 (3)
DMA-DT	22 ± 2.4 (3)	28 ± 0.27 (3)	41.4 ± 4.4 (3)
TMA-DT	3.7 ± 0.39 (3)	20 ± 2.0 (3)	18.0 ± 2.8 (3)

Abbreviations: *Ls* BP = *Lymnaea stagnalis* AChBP; *Ac* BP = *Aplysia californica* BP; *Ac* BP Y55W = *A. californica* BPY55W mutant.

## Data Availability

Data from experiments summarized in the figures and tables are available from W.K. on request.

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
