# Peer review of "Interactions of Nereistoxin and Its Analogs with Vertebrate Nicotinic Acetylcholine Receptors and Molluscan ACh Binding Proteins"

_marinedrugs, 2022, doi:10.3390/md20010049_

Round 1

Reviewer 1 Report

In this manuscript, Kem et al have assessed for the first time the binding ability of nereistoxin (NTX) to two major brain neuronal nAChRs (alpha4beta2 and alpha7). This toxin has been previously shown to act as non-competitive antagonist (probably channel blocker) at insect nAChRs (effective insecticide) and also as an antagonist at fish and mammalian muscle nAChRs, probably acting at their ACh-binding site. The authors have used rat brain preparations and Torpedo electric organ membranes (rich in muscle-type nAChRs) and assessed the ability of NTX and analogs to compete with radio-labelled classical ligands (cytisine for a4b2 and a-BGTX for alpha7 and muscle nAChRs) for binding to the respective receptors apparent in these issues. Authors clearly demonstrate the inhibition of radioligand binding to the two major neuronal nAChRs and to Torpedo muscle nAChRs, with the a4b2 subtype showing the highest affinity for NTX (in the low micromolar range). They further used various methylated analogs in order to disrupt the disulfide bond of the NTX molecule (DMA-DT and TMA-DT) and also yield quaternary analogs (Met-NTX and TMA-DT). The radio-ligand binding competition experiments performed with these analogs showed that NTX action does not require disulfide bond interchange between the NTX and the highly conserved loop C vicinal disulfide bond since TMA-DT showed increased affinity to alpha7 and muscle nAChRs and DMA-DT showed only a less than 3-fold decreased affinity for the alpha4beta2 nAChR. These results are also strengthened by use of a molluscan homologous AChBP lacking the pair of bonded cysteins at its loop C. Further, authors demonstrate that quaternization of the amine nitrogens of both NTX and DMA-DT is the most critical factor leading to an increase in the affinity of MaNTX and TMA-DT to at least alpha7 and muscle nAChRs. Overall, authors clearly demonstrated the competitive antagonistic character of such a small molecule (NTX) and propose that use of labelled MetNTX can be a very useful probe to detect nAChRs in tissue preparations and also for any toxin adducts.

Minor points:

- Authors should provide the structures of all analogs for better interpretation by readers with not adequate chemical background.

- Paragraph 2.4: Please discuss the AChBPs used in this paragraph (not in the legend of Figure 7) and point out the main difference apparent in AChBP-3; also

- Also, in the Discussion, I would strongly suggest that authors insert a paragraph to describe in a simpler way the effect of the disulfide bond of the original NTX on the binding ability of the analogs. For example, readers may be confused by the conclusion that impairment of the disulfide bond of NTX is not deleterious for binding to nAChRs, since in the case of DMA-DT the binding ability to alpha7 and Torpedo nAChRs is practically abolished (while retained in the case of alpha4beta2). Please explain better.

- Authors should review the inserted boxes in Figures 2, 5, 6 and 7. Since, as denoted in the Discussion, authors would not like to use the Ki values, these should not be included in the Figures. Also, remove the dates of exps.

- Lines 30-31: Rephrase the sentence, since TMA-DT showed decreased affinity compared only to Me-NTX in the cases of alpha7 and Torpedo, whereas DMA-DT showed decreased affinity to both NTX and Me-NTX in all cases.

- Lines 33-36: The reader cannot understand the conclusion why the disulfide bond of NTX is not required for binding to AChBPs; please explain better (use of AChBP-3 mutant)

- Line 108-109: correct to inhibition equilibrium constants

-  Line 126: Rewrite to “Effect on alpha7 neuronal receptors

- Line 168: replace (2) with (3)

- Line 291-292: Do you mean Ki ? Also, consider leaving only the IC50 values as discussed above.

Overall, the manuscript by Kem et al. merits publication to Marine Drugs after some improvements in order the conclusios drawn to be better comprehended by the average reader.

Author Response

Your comments were appreciated and the ms was revised accordingly. An additional analog structure figure (2) is incorporated. Inserted boxes in figures containing unnecessary information are made invisible. Ki values are removed. Ambiguous statements about analog relative affinities are eliminated. The conclusion that the disulfide bond of NTX is unnecessary is based on similar affinities of the dithiane analogs lacking the disulfide to NTX and MeNTX, and the lack of preincubation effect on epibatidine binding to AChBPs. Additional data on AChBPs is incorporated into a new table (2).

Reviewer 2 Report

The study by Kem et al. aims to investigate the interaction of nereistoxin (NTX) and its analogs with some vertebrate nicotinic acetylcholine receptors (nAChRs), as well as molluscan acetylcholine-binding proteins (AChBPs). The authors intended to find out the structural bases of antagonistic activity towards these receptors for such a small molecule as NTX. To do this, they synthesized three analogs containing a broken S-S bond in dithiolane cycle and/or modification into a quaternary ammonium cation. The study was carried out on neuronal rat α4β2 and α7 nAChRs and muscle-type receptor from Torpedo ray electric organ, as well as AChBP from Limnea stagnalis and Aplysia californica in natural and mutant form, using radioligand and FlexStation functional assays. The authors showed that breaking the S-S bond in the cycle slightly decreases the affinity to all receptors, and obtaining the cationic form of NTX resulted in a similar effect only for the α4β2 receptor, while significantly increasing the affinity for the α7 and the muscle-type nAChRs (as it was earlier demonstrated for nicotine). The authors concluded that the disulfide bond of NTX is not required to react with the loop C vicinal disulfide in the AChBP and nAChRs and consider it noteworthy to further clarify the molecular basis of NTX antagonism.

When reading the manuscript, I had several wishes to improve the perception of the presented results –

  1. It would be reasonable to collect all the calculated inhibition parameters for all targets in Table 1, where, in addition to the IC50 values, it should also indicate the Hillslopes. It seems also reasonable to add to this Table the results obtained on the AChBPs, and expressed in the same way (IC50 values and Hillslopes), so as not to look for them in the Figures. Accordingly, these calculated parameters, as well as headings, can be removed from the field of Figures.
  2. There is no explanation why experiments using FlexStation functional assays were carried out on human α4β2 nAChR, and not a rat one as in radioligand analysis. A large number of species-selective ligands are known, therefore, the correlation of the results obtained by two methods on the different α4β2 receptors will not be entirely correct.
  3. In Figure 1, the formulas and the obtained analogs of NTX can be given for visual clarity.
  4. Provide a reference (or briefly describe) the obtaining of AChBPs, including the mutant form.
  5. Minor typos noticed - line 126 - move the dot in the title; lines 127-130 - remove italics; line 168 – (3).

Author Response

The referee comments were appreciated and the ms has been revised accordingly. An additional table (2) was added to present the AChBP binding data. The data is more complete than in the original ms. The Hill slopes were not included because they were not informative (they didn't vary with the analog) and were in the range expected from previous studies (citation 12, Xing et al., 2020). We used cells expressing the human alpha4beta2 receptor because cells expressing the rat receptor are unavailable. An additional figure (2) is added to show the structures of the other nereistoxin analogs. An additional refeerence (Hansen et al., 2002) is given for source of the AChBPs.